# Validation of Six Commercial Antibodies for the Detection of Heterologous and Endogenous TRPM8 Ion Channel Expression

**DOI:** 10.3390/ijms232416164

**Published:** 2022-12-18

**Authors:** Pablo Hernández-Ortego, Remedios Torres-Montero, Elvira de la Peña, Félix Viana, Jorge Fernández-Trillo

**Affiliations:** Instituto de Neurociencias (IN), CSIC-UMH, 03550 San Juan de Alicante, Spain

**Keywords:** TRPM8, Western blot, immunocytochemistry, immunohistochemistry

## Abstract

TRPM8 is a non-selective cation channel expressed in primary sensory neurons and other tissues, including the prostate and urothelium. Its participation in different physiological and pathological processes such as thermoregulation, pain, itch, inflammation and cancer has been widely described, making it a promising target for therapeutic approaches. The detection and quantification of TRPM8 seems crucial for advancing the knowledge of the mechanisms underlying its role in these pathophysiological conditions. Antibody-based techniques are commonly used for protein detection and quantification, although their performance with many ion channels, including TRPM8, is suboptimal. Thus, the search for reliable antibodies is of utmost importance. In this study, we characterized the performance of six TRPM8 commercial antibodies in three immunodetection techniques: Western blot, immunocytochemistry and immunohistochemistry. Different outcomes were obtained for the tested antibodies; two of them proved to be successful in detecting TRPM8 in the three approaches while, in the conditions tested, the other four were acceptable only for specific techniques. Considering our results, we offer some insight into the usefulness of these antibodies for the detection of TRPM8 depending on the methodology of choice.

## 1. Introduction

The detection of environmental temperatures is one of the critical functions of the somatosensory system. Transient Receptor Potential Melastatin 8 (TRPM8) is a non-selective cation channel that permeates divalent (Ca^2+^) and monovalent (Na^+^, K^+^) cations [1,2]. It is a polymodal channel that can integrate different physical and chemical stimuli, being activated by mild cold and different natural and synthetic cooling compounds such as menthol, WS-12 [3] and icilin [4]. In addition, its activity is modulated by intracellular signaling molecules such as Ca^2+^ and PIP_2_ [5,6,7].

TRPM8 is preferentially expressed in Aɗ and C-fibers innervating the skin where it acts as a mild cold temperature transducer [8]. The depolarization of cold thermoreceptor endings generates trains of action potentials that travel from the periphery towards the cell body in the dorsal root (DRG) or trigeminal ganglia (TG), connecting with dorsal horn interneurons and to other brain regions (e.g., hypothalamus, cortex) where ultimately cold perception occurs and behavioral and autonomic thermoregulatory responses are triggered. 

TRPM8 is also expressed in other surface tissues such as the cornea, where it participates in the regulation of the humidity of the eye surface by the detection of the tear osmolality. The activation of corneal TRPM8+ endings triggers basal tearing and blinking, and its malfunction is implicated in the mechanisms of dry eye disease (DED) [9,10,11,12,13].

The expression of TRPM8 has also been described in tissues that are not exposed to the environment such as the brown adipose tissue where it has a role in thermogenesis and high-fat diet-induced obesity [14], intestinal epithelium associated with irritable bowel syndrome (IBS) and colitis [15,16,17], and the bladder associated with cooling-reflex, urinary urgency, overactive bladder and painful bladder syndrome [18,19,20,21]. Recently, TRPM8 expression has been detected in some brain regions (hypothalamus, septum, thalamus) and the retina, suggesting a role of this channel in thermal regulation and circadian control [22,23]. Additionally, TRPM8 is also expressed in prostate cancer cells. Some studies suggest a role of this channel in cell proliferation, whereas other findings suggest its participation in the reduction in metastatic processes in the prostate [24].

The important physiological role of TRPM8, as well as its involvement in different pathophysiological conditions (prostate cancer, migraine, obesity, cold pain, itch, inflammation), makes this channel an important target for different studies. Currently, multiple techniques are routinely used to assess TRPM8 function. They include electrophysiological recordings and calcium imaging. Various mouse models are also available, including reporter and KO mice. In particular, reporter mice have been extremely useful in characterizing the expression pattern of TRPM8 in the mouse peripheral and central nervous system [22,25,26]. These studies, together with previous in situ hybridization techniques [27], indicate that TRPM8 is expressed in a small subpopulation of adult TG and DRG sensory neurons, and shows a restricted expression in the brain as well. In other species, the lack of reporter animals obliges the use of alternative techniques, such as immunofluorescence and Western blot (WB), to quantify TRPM8 expression.

However, while immunofluorescence and WB are antibody-based mainstream techniques for protein visualization and quantification, the use of antibodies to characterize the expression of TRPM8 and other TRP channels remains problematic [28]. There are many commercially available antibodies but their performance in different techniques such as WB, immunocytochemistry (ICC) and immunohistochemistry (IHC) varies from acceptable to very poor, with many antibodies showing low specificity. To our knowledge, no systematic profiling of commercial antibodies against TRPM8 has been reported so far. Here, we validated six commercial TRPM8 antibodies for their use in WB, ICC and IHC, using different methodologies and following standard procedures. First, mouse TRPM8 fused to the fluorescent protein EYFP (mTRPM8-EYFP) was expressed in HEK-293 cells. ICC was performed in fixed cells testing the performance of the antibodies under TRPM8 overexpression conditions. In parallel, cell lysates were subjected to WB analysis with each of the tested antibodies using manufacturer-recommended dilutions. Finally, the antibodies were tested against native TRPM8, using DRG sensory neurons from a transgenic mouse line that specifically labels TRPM8+ neurons with EYFP. To confirm their specificity, the antibodies that performed best in DRG ICC and IHC were further validated using a TRPM8 KO mouse.

## 2. Results

Six commercially available antibodies against TRPM8 were characterized in this study and summarized in Table 2 in the Material and Methods Section. At present, five of them are still available, and one has been discontinued (Origene1). All antibodies were designed against human TRPM8 but have reported cross-species reactivity with mouse and rat. As detailed in Table 2, two antibodies were polyclonal and four were monoclonal. The majority are directed against the extracellular domains of TRPM8 (although the exact region was only provided for Alomone) while Origene2 epitope is located in the N-terminal domain.

### 2.1. All Tested Antibodies Specifically Label Mouse TRPM8 Overexpressed in HEK-293 Cells

Cultured HEK-293 cells transfected with mTRPM8-EYFP were used to assess the suitability of each antibody in detecting overexpressed mTRPM8 by immunocytochemistry (ICC). EYFP fluorescence was used to identify transfected cells expressing TRPM8. First of all, using calcium imaging, we confirmed that EYFP+ HEK-293 cells responded to TRPM8 agonists such as mild cold stimuli (∼18–20 °C) and 10 µM WS-12, while EYFP− cells did not (Figure 1). Only cells that responded to 50 µM Carbachol, a compound that activates endogenous muscarinic receptors in HEK-293 cells [29], were included in the analysis.

For mTRPM8 detection in ICC, we used two dilutions of each antibody in the range recommended by the vendors (1:500 or 1:200). All antibodies showed a clear co-localization of EYFP and mTRPM8 immunostaining (Figure 2A–F), indicating that the six antibodies are successfully detecting high levels of mTRPM8 protein. Notice the fluorescence punctate pattern typical of TRPM8 overexpression [30,31,32], both for the EYFP and antibody signal, further confirming antibody specificity (Appendix A). No TRPM8 immunostaining was found in neighboring EYFP− (i.e., non-transfected) cells, showing that none of the antibodies displayed any confounding unspecific staining. To compare the performance of the different antibodies, a specificity ratio (SR) (see Methods) was calculated for each dilution (Figure 2A–F, box plots). In general, the SR was similar with both dilutions or higher with 1:200 (ECM2 and Origene2), with the exception of ECM1, in which the higher dilution (1:500) showed a higher SR. In other words, ECM1 seems to work better when more diluted.

As shown in Figure 2G, the six antibodies have an SR significantly higher than the corresponding control without primary antibody (SR > 1, meaning that the antibody fluorescence signal is higher for TRPM8+ than for TRPM8− cells). However, SRs calculated using ECM1 and Origene1 were almost two-fold higher than with the other antibodies.

### 2.2. Not All Antibodies Detect mTRPM8 in Western Blotting

Cell lysates from HEK-293 cells transfected with mTRPM8-EYFP were used to examine whether the mentioned antibodies were able to detect mTRPM8 in Western blot (WB) (Figure 3A–F). A band with a molecular size around 160 kDa (black arrowheads), corresponding to the expected size of mTRPM8-EYFP, was detected by ECM1, Origene1 and Origene2. However, this band was not detected by Alomone, ECM2 or ECM3, even though EYFP immunoblotting revealed the presence of mTRPM8-EYFP in the sample (green arrowheads).

### 2.3. ECM1 and Origene1 Specifically Stain Endogenous TRPM8 in Mouse Dorsal Root Ganglion Sensory Neurons

The specificity of the six antibodies against endogenously expressed TRPM8 was assessed by ICC and immunohistochemistry (IHC) in mouse dorsal root ganglion (DRG) neurons. These cells are first order sensory neurons that respond to diverse stimuli and include a subpopulation expressing TRPM8 that are activated by mild-cold and WS-12. In order to identify TRPM8+ neurons, we used a transgenic mouse line that expresses EYFP under the TRPM8 promoter (*Trpm8^BAC^-EYFP^+^*). Note that in this case, EYFP is distributed freely in the cytoplasm and is not indicative of TRPM8 subcellular localization (Appendix A). Previous functional studies [33] indicated that, in this mouse line, EYFP expression is a good proxy of TRPM8 expression. Here, we confirmed the suitability of this reporter mouse to identify TRPM8-expressing neurons. Using cellular calcium imaging (Figure 4A,B), we found that the majority of cultured DRG cells with a strong EYFP signal responded to mild cold (∼18–20 °C) and 10 µM WS-12 (100 % and 92.3 %, respectively) while only a few non- or dimly fluorescent cells responded (2.37 % and 1.05 %). Only cells responding to high KCl (30 mM) were analyzed.

All six antibodies were tested for ICC on cultured DRG sensory neurons from *Trpm8^BAC^-EYFP^+^* mice. Two recommended antibody dilutions (1:500 and 1:200) were used. Only two of the six antibodies performed well; the ECM1 and Origene1 antibodies stained specifically the EYFP+ cold responsive cells (Figure 4C and Figure 5B,C), showing no clear signal in EYFP− neurons, and displayed an SR > 1 (Figure 5B,C,G). In our conditions, the other four antibodies performed poorly. Alomone stained almost all sensory neurons (whether EYFP+ or EYFP−, Figure 5A), which is inconsistent with the restricted expression of TRPM8. In contrast, ECM2 did not label any cell (Figure 5D). ECM3 stained some cells, but it was not specific for EYFP+ neurons (Figure 5E). Origene2 seemed to mark EYFP+ cells with more intensity than EYFP− but showed some unspecific staining as well (Figure 5F).

The antibody concentrations used did not seem to have an important effect in their SR, with the exception of Alomone and ECM1, which showed a higher SR with 1:500 dilution (Figure 5A–F, box plots). Comparing the SR of each antibody versus the control (no primary antibody) (Figure 5G), we observed that ECM1 and Origene1 have a substantially higher ratio, suggesting good specificity. Origene2 SR is slightly higher than the control, indicating that this antibody also shows some specificity for TRPM8 in the conditions used.

Endogenous protein expression is more commonly assessed with IHC than with ICC, as the sample is more easily acquired and the tissue structure is preserved. However, other challenges arise such as epitope accessibility and masking [34,35]. We tested all six antibodies in the IHC of mouse DRG slices. Initially, citrate antigen retrieval was used, but neither antibody worked properly; subsequent IHCs were performed in the absence of antigen retrieval. The same two dilutions (1:500 and 1:200) were used, without any significant difference in the SR between them (Figure 6A–F, box plots). Similar to the ICCs, only ECM1 and Origene1 showed staining restricted to the bright EYFP+ neurons and distinguishable from the unspecific signal (Figure 6B,C). Alike the ICC results, Alomone marked all cells unspecifically (Figure 6A), and ECM2 and ECM3 produced an unspecific staining pattern similar to the staining observed in the secondary antibody alone (Figure 6D,E and Appendix A). In addition, some EYFP− cells were strongly labelled after ECM2 incubation. With Origene2, the EYFP+ cells were immunostained, but a fraction of the EYFP− cells was also unspecifically labeled. 

The SR comparison among the different antibodies is shown in Figure 6G: only ECM1 (1:200 dilution) and Origene1 have an SR different from the control without primary antibody. 

### 2.4. KO Validation of TRPM8 Antibody Specificity in KO Animals

Knockout cells provide the best negative control in antibody validation [36]. Both the ICC and IHC results suggest that only the ECM1 and Origene1 antibodies perform well in detecting the endogenous expression of mTRPM8. To validate the specificity of these two antibodies, both ICC and IHC were repeated using DRG-cultured cells and slices from TRPM8 KO mice (*Trpm8^EGFPf^*; B6;129S1(FVB)-*Trpm8^tm1Apat^*/J) [25] that express farnesylated EGFP instead of TRPM8 (i.e., EGFP marks neurons that should have expressed TRPM8). Using either ECM1 or Origene1, specific antibody fluorescence was absent in the EGFP+ cells of the TRPM8 KO mouse (Figure 7 and Appendix A). An unspecific signal was detected in some EGFP- cells (Figure 7A–D), albeit with a different fluorescence distribution and dimmer intensity than in EYFP+ cells in the TRPM8 reporter mouse (*Trpm8^BAC^-EYFP^+^*) (Appendix A). Furthermore, the SR for both antibodies was significantly higher in the reporter mouse compared to the TRPM8 KO, both in ICC and IHC (Figure 7E,F).

## 3. Discussion

The use of antibodies to quantify the level or expression pattern of antigens (e.g., proteins) is one of the most popular and valuable techniques in cell biology. It is common knowledge that the performance of many antibodies is substandard, with low sensitivity and poor specificity [36]. Unfortunately, very often, negative results are not reported in the literature. In other cases, antibody specificity is not assessed critically, which could lead to the erroneous interpretation of the results. Thus, the proper characterization of antibodies is of utmost importance [36,37].

Our main aim was to compare the performance of six commercial TRPM8 antibodies in Western blot (WB), immunocytochemistry (ICC) and immunohistochemistry (IHC). Our results provide useful hints for choosing the most adequate antibodies for different TRPM8 protein detection techniques. In our hands, the six antibodies analyzed show diverse suitability for each technique and expression system. According to the respective manufacturers, all six antibodies should work for WB. However, using their recommended dilutions, only three of them yielded satisfactory results for our standards. Interestingly, the Alomone antibody has been widely used for WB [38,39,40], but in our conditions, it did not work properly. However, in many of these articles, the antibody dilution was lower than recommended, and the reported TRPM8 bands were dubious and/or no full membrane/molecular weight markers were shown, making it difficult to rule out the presence of unspecific bands or cross-detection. In some cases, the reference to the antibody used was not clear; we could not rule out that it was a different antibody from the one we tested. Additionally, some studies targeted human TRPM8 [41,42], and we used the mouse ortholog instead. The Alomone antibody immunogen is publicly known (Table 2); it comprises 13 residues of human TRPM8, from which two are different in human and mouse. This may explain the discrepancy between the results of this work and what others reported with human TRPM8. ECM2 and ECM3 also failed to produce a good WB signal. In one of the replicates with ECM2, we detected a faint band of a molecular weight compatible with TRPM8, but it is also present in the lysate from non-transfected cells, thus questioning its specificity. Many other unspecific bands of different molecular weights were also detected (data not shown), so special caution has to be taken if using this antibody for WB. At the time of preparing this manuscript, to our knowledge, no published work references either ECM2 or ECM3 for WB. Interestingly, both ECM2 and ECM3, as well as Alomone, worked well in the ICC of mTRPM8 transfected HEK-293 cells, suggesting that they are in fact able to bind to mTRPM8. The discrepancy observed between WB and ICC could be explained by the fact that cell lysis and denaturing conditions used in WB may make the epitope less accessible to the antibody paratope. For those reasons, a potential use of these three antibodies for WB after improving conditions cannot be excluded.

ECM1, Origene1 and Origene2 produced consistently good results for WB and ICC in a TRPM8 overexpression system, indicating that, at least in the conditions of this study, they can be confidently used for both techniques. Limited literature references Origene1 [43]. No previous WB records have been found for ECM1 or Origene2, making this study the first to show their appropriateness for WB. 

Proteins are differentially expressed in heterologous and endogenous systems, being generally more abundant in the former. Endogenous TRPM8 detection was more challenging, and only ECM1 and Origene1 worked well both in dorsal root ganglion (DRG) ICC and IHC. We cannot give a firm judgement about Origene2, as the immunostaining in DRG ICC was ambiguous. An additional consideration with Origene2 is that it was raised in mouse, and thus the anti-mouse secondary antibody could bind to immunoglobulins present in the mouse DRG, distorting the interpretation of the results (Appendix A). Testing this antibody in human samples could help in resolving this issue. These results, together with its good performance in the WB and ICC of HEK-293 cells, suggest that Origene2 could potentially be a good antibody for endogenous TRPM8 detection after adjusting experimental protocols.

In our hands, Alomone, ECM2 and ECM3 did not have enough sensitivity and/or specificity to detect endogenous TRPM8, at least with the conventional conditions we used. There is abundant literature using Alomone antibody for the detection of endogenous TRPM8 by ICC and/or IHC in different tissues [39,44,45,46,47,48]. However, in many of the published figures, the staining pattern was not consistent with the known restricted expression of TRPM8 [31,49]. In many cases, all cells seem to be stained with fluorescence evenly spread across the entire cell. In contrast, the fluorescence we see with antibodies that we consider good (ECM1 and Origene1) is distributed in a punctate pattern (Appendix A). Moreover, in some studies, this antibody marks almost all cells subjected to immunochemistry. This is inconsistent with the known expression pattern of TRPM8 in DRG neurons, which is restricted to a small subpopulation of about 10–15 % cells. In light of our results, we believe that caution should be taken interpreting the immunofluorescence signal from those studies. Similar to the reasons proposed for WB, we cannot exclude that this antibody could perform better in human tissue or in different experimental conditions.

Our data lead us to suggest the use of ECM1, Origene1 and Origene2 for WB and ECM1 and Origene1 for the immunochemistry of endogenous TRPM8. A brief comparison of the performance of the tested antibodies with the different methodologies is shown in Table 1.

Origene1 has been discontinued. According to the available information, it seems to come from the same clone (EPR4196(2)) as Abcam ab109308 (also discontinued) that has been used with good results for WB and IHC [49,50,51]. We think that Abcam antibody could still be available in many labs and could be a good alternative for TRPM8 detection.

### Limitations of the Study

Variations in immunodetection protocols result in countless procedures in the literature. The variables include fixation, sample preparation and tissue processing, antibody concentration, antigen retrieval, permeabilization, the blocking of non-specific sites, signal amplification, detection method and others [52]. There is no generic protocol for immunolabeling. We standardized the labeling procedure for the six antibodies tested but did not perform an in-depth exploration of the protocol space. It is perfectly feasible that other conditions would have resulted in better results with some of the antibodies tested. Various online resources provide useful tips on step-by-step guides for staining optimization.

## 4. Materials and Methods

### 4.1. Animals

All experimental procedures were performed in accordance with the Spanish Royal Decree 53/2013 and the European Community Council Directive 2010/63/EU. Adult mice (2–4 months old) of either sex were used. The mice were housed in a temperature-controlled room (21 °C) on a 12 h light/dark cycle, with access to food and water ad libitum.

#### Mouse Lines

*Trpm8^BAC^-EYFP^+^* BAC transgenic line was generated in our laboratory: enhanced yellow fluorescent protein (EYFP) is expressed under the promoter of TRPM8 [33].

*Trpm8^EGFPf^;* B6;129S1(FVB)-*Trpm8^tm1Apat^*/J was obtained from Ardem Patapoutian (Scripps Research Institute) and expresses enhanced green fluorescent protein (EGFP) after the TRPM8 start codon; homozygous mice are null for TRPM8 [25]. The lox-P flanked neomycin cassette introduced in the Trpm8 locus during the generation of the transgene was removed to enhance GFP expression [53]. The genotype of all mice was confirmed by PCR.

### 4.2. Cell Line Culture and Transfection

The HEK-293 cell line was obtained from ECACC (Salisbury, UK). Cells were maintained at 37 °C, 5% CO2 in DMEM medium (Thermo Fisher Scientific, Waltham, MA, USA) supplemented with 10% fetal bovine serum and 1% penicillin/streptomycin. Twenty-four hours before transfection, the cells were seeded on 6-well plates. For ICC, the wells were filled with poly-L-lysine (0.01%, Sigma-Aldrich, St. Louis, MO, USA) treated 6 mm diameter #0 glass coverslips. The cells were transfected with mTRPM8-EYFP (mTRPM8 fused with EYFP at its C-terminus) [54] using a LipofectamineTM 2000 (Thermo Fisher Scientific). In total, 3 μg DNA and 3 μL Lipofectamine in 300 µL Optimem (Thermo Fisher Scientific) were used per well containing 700 µL DMEM. Then, 24–48 h post-transfection, mTRPM8-EYFP expression was monitored with an epifluorescence microscope, and the cells were either lysated (for WB) or fixed in 4% PFA (for ICC).

### 4.3. Mouse DRG Extraction and Neuronal Culture

Mouse DRG neurons were extracted and dissociated as previously described [55]. Briefly, the mice were euthanized by cervical dislocation. The spinal cord was removed and 20–40 DRGs were dissected and washed in cold HBSS solution (Thermo Fisher Scientific). For IHC, whole ganglia were directly fixed for 2 h in 4 % PFA. For ICC, ganglia were then incubated in 900 U/mL type XI collagenase (Sigma-Aldrich) and 5.46 U/mL dispase (Thermo Fisher Scientific) for 45 min at 37 °C in 5% CO_2_. After enzymatic treatment, the ganglia were mechanically dissociated using fire-polished glass pipettes in calcium-free solution containing HBSS (Thermo Fisher Scientific), 1% MEM-Vit (Thermo Fisher Scientific), 10% fetal bovine serum (Thermo Fisher Scientific) and 100 mg/mL penicillin/streptomycin. The cell suspension was centrifuged, and the pellet was resuspended in culture medium containing: MEM (Thermo Fisher Scientific), 1% MEM-Vit (Thermo Fisher Scientific), 10% fetal bovine serum (Thermo Fisher Scientific) and 100 mg/mL penicillin/streptomycin. The cells were then seeded onto 6 mm diameter glass coverslips previously coated with 0.01% poly-L-lysine (Sigma-Aldrich, St. Louis). Twenty-four hours after seeding, the cells were subjected to calcium imaging or fixed for 10 min in 4%PFA.

### 4.4. Calcium Microfluorometry

A Fura2-AM calcium indicator (Thermo Fisher Scientific) was used for ratiometric calcium imaging experiments. The cells were loaded with 5 μM Fura2-AM and 400 ng/mL Pluronic F-127 (Thermo Fisher Scientific) in control external solution for 45 min at 37 °C in 5% CO2. The coverslips containing the cells were placed in a low volume chamber mounted on an inverted microscope (Leica DMI3000B, Leica Microsystems, Wetzlar, Germany) and continuously perfused with fresh solution at a rate of ~1ml/min. The Fura2-AM was excited at 340 and 380 nm with a Lambda 10-2 filter wheel and a Lambda LS xenon arc lamp (Sutter Instruments, Novato, CA, USA). Emission fluorescence was filtered with a 510 nm long-pass filter. Images were acquired with an Orca ER CCD camera (Hamamatsu Photonics, Hamamatsu, Japan) at a frequency of 0.33 Hz and analyzed with MetaFluor software (Molecular Devices, San Jose, CA, USA). Cytosolic calcium increases are presented as the ratio of emission intensities after sequential excitation at 340 and 380 nm (F340/F380). Measurements of the HEK293-cells and DRG neurons were performed at 32–34 °C using a homemade water-cooled Peltier system controlled by a temperature feedback device.

### 4.5. Western Blot

All WB experiments were performed on samples obtained from HEK-293 cells transfected with mTRPM8-EYFP. The cells were harvested 48 h after transfection, centrifuged at 800× *g* 10 min and washed with cold PBS twice. The pellets were lysed (Buffer Lysis: 50 mM Tris-HCl, pH 7.5, 120 mM NaCl, 0.5 mM EDTA, 0.5 % Nonidet P-40 with phosphatase, and protease inhibitors (cOmplete Mini, Roche)). The cell lysates were sonicated for 10 min at 4 °C and centrifuged at 10,000× *g* for 15 min at 4 °C. The protein amount was quantified (Pierce TM BCA Protein Assay Kit, (Thermo Fisher Scientific), and the samples were diluted with loading buffer (10% sodium dodecyl sulfate, 312.5 mM Tris-HCl pH 6.8, 50% glycerol and 0.05% bromophenol blue). The protein samples were subjected to SDS-PAGE (15 µg per lane) and blotted onto Protran Nitrocellulose Membrane—Whatman (GE Healthcare Life Science, Chicago, IL, USA). The membranes were blocked with blocking solution (5% powdered milk in TBST: Tris-buffered saline with 0.05% Tween-20) and incubated overnight at 4 °C with the respective anti-TRPM8 primary antibodies (see Table 1). Subsequently, membranes were washed with TBST, incubated with horseradish peroxidase (HRP) conjugated secondary antibody and developed with ECLplus (Thermo Fisher Scientific) and imaged in an Amersham Imager 680 device. After several washes with TBST, membranes were incubated overnight at 4 °C with anti-GFP (1:1000) and anti- GAPDH (1:5000) antibodies, developed and imaged as explained before.

### 4.6. Immunocytochemistry

Cells, cultured on coverslips, were fixed 10 min in 4% PFA in 0.1M PBS, washed three times in PBS and twice in TTBS (0.5 M Tris Base, 9% w/v NaCl, 0.5% Tween 20, pH 7.6) for 10 min. Non-specific binding sites were blocked by incubating the cells for 30 min in freshly prepared blocking solution (1X TTBS, 1% bovine serum albumin (Tocris Bioscience, Bristol, UK) and 0.25% Triton-X100). The cells were then incubated for 2 h at RT with the primary antibodies. Afterwards, the cells were washed three times with TTBS 10 min and incubated for 45 min at RT with the secondary antibodies. Both the primary and secondary antibodies were diluted in blocking solution. The cells were washed again three times with TTBS. An extra step was added for HEK cells that were incubated for 5 min in Hoechst 33342 (Thermo Fisher Scientific). Then, the cells were washed with PBS and once with ddH2O and mounted on a microscope slide using a VectaShield H-1000 antifade mounting medium (Vector Laboratories, Burlingame, CA, USA). Mounted coverslips were sealed with clear nail polish and stored at 4 °C until imaging. Z-stack images with a step size of ∼ 2 µm were acquired with an UPlanSApo 20× or a PlanApoN 60× objective using an inverted confocal microscope (Olympus FV1200) driven by FV10-ASW 4.2 software (Olympus Life Sciences, Waltham, MA, USA).

### 4.7. Immunohistochemistry

Whole DRGs were fixed for 1 h in 4% PFA in 0.1 M PBS, washed twice in PBS for 10 min and cryoprotected in 30% sucrose overnight at 4 °C. The next day, tissue was embedded in an optimal cutting temperature compound (OCT Tissue-Tek, Sakura Finetek, Torrance, CA, USA) and frozen in dry ice. Then, 20 μm slices were cut using an MNT cryostat (Slee Medical, Nieder-Olm, Germany) and placed on SuperFrost microscope slides (Thermo Fisher Scientific). The slides were dried in an oven at 37 °C for 30 min and washed twice with PBT (0.1 M PB, 0.05% Tween20, pH 7.4). Antigen retrieval (when used) was performed by boiling the samples in pH 6.0 citrate buffer for 20 min using a water bath and letting them cool down in the bath for 10 min. Non-specific binding sites were blocked for 1 h with a blocking solution containing 5% bovine serum albumin and 1% Triton X-100 in PBT. The slides were incubated with the primary antibodies overnight at 4 °C. The next day, the slides were washed four times with PBT 10 min each and incubated with the secondary antibodies for 2h at RT. Both the primary and secondary antibodies were diluted as shown in Table 2 in blocking solution. Then, the slides were washed four times with PBT 10 min, once with PBS for 10 min and once with ddH_2_O for 5 min. Finally, the slides were dried in dark conditions at RT, and a glass coverslip was mounted on top of each slide using Fluoromount mounting medium (Sigma-Aldrich). Z-stack images with a step size of ∼2 µm were acquired with an UPlanSApo 20× or a PlanApoN 60x objective using an inverted confocal microscope (Olympus FV1200) driven by FV10-ASW 4.2 software (Olympus Life Sciences).

### 4.8. Image Analysis and Quantification

All quantitative measurements were performed using the 16-bit raw maximum projection images without any further modification. We defined the antibody specificity ratio (SR) in a TRPM8+ cell as the mean signal intensity of the antibody fluorescence in that cell divided by the average of the mean signal intensity in several (at least 20) TRPM8− cells in the same field. An SR value of 1 indicates the same antibody fluorescence intensity in TRPM8+ and TRPM8− cells, while an SR > 1 indicates higher fluorescence intensity in TRPM8+ than in TRPM8− cells. TRPM8+ cells were identified by EYFP immunofluorescence, either in transfected cells or using a reporter mouse line. The images were analyzed with ImageJ 1.51j8 (NIH, Bethesda, MD, USA) and Origin 2019 (OriginLab, Northampton, MA, USA).

### 4.9. Image Display

The immunofluorescence image brightness and contrast were only adjusted for image presentation to enhance visualization without altering any other feature. In the case of the TRPM8 antibody signal images, the maximum red pseudocolor level was set for the maximum fluorescence intensity; thus, images displaying high background noise correspond to experiments in which the antibody signal was poor.

### 4.10. Statistical Analysis

The values are given as indicated in each figure caption. The normality of data distribution was checked with Shapiro–Wilk and Kolmogorov–Smirnov tests. Statistical significance was estimated with the Mann–Whitney test or Kruskal–Wallis followed by Dunn’s post hoc test. A *p*-value < 0.05 was considered statistically significant. In the case of small samples (n < 100), the significance was curtailed to * *p* < 0.05 and ** *p* < 0.01. The analysis was performed using Prism version 8 (GraphPad Software, San Diego, CA, USA). 

### 4.11. Antibodies Used in This Study

All primary and secondary antibodies used are summarized in Table 2.

## Figures and Tables

**Figure 1 ijms-23-16164-f001:**
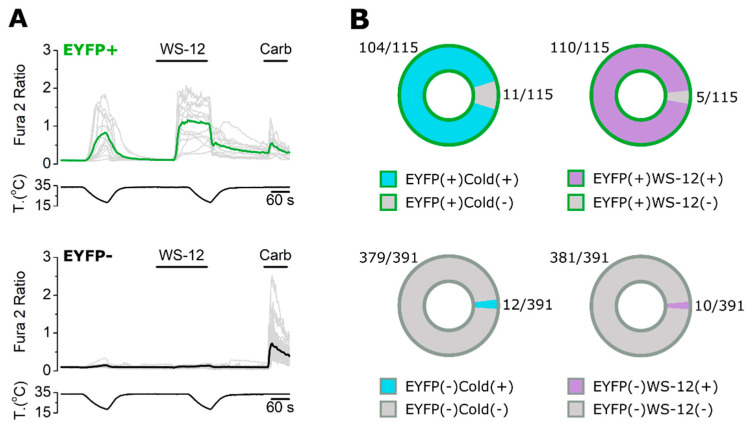
Calcium imaging in HEK-293 cells transiently overexpressing mTRPM8-EYFP. (**A**) Representative traces of calcium transients evoked by mild cold (∼18–20 °C), WS-12 (10 µM) and Carbachol (50 µM) in both EYFP+ and EYFP− cells. Traces corresponding to individual cells are shown in grey. Colored traces represent averages of the respective individual traces. (**B**) Proportion of EYFP+ and EYFP− cells responding to cold and WS-12. n = 506 cells; 2 independent experiments.

**Figure 2 ijms-23-16164-f002:**
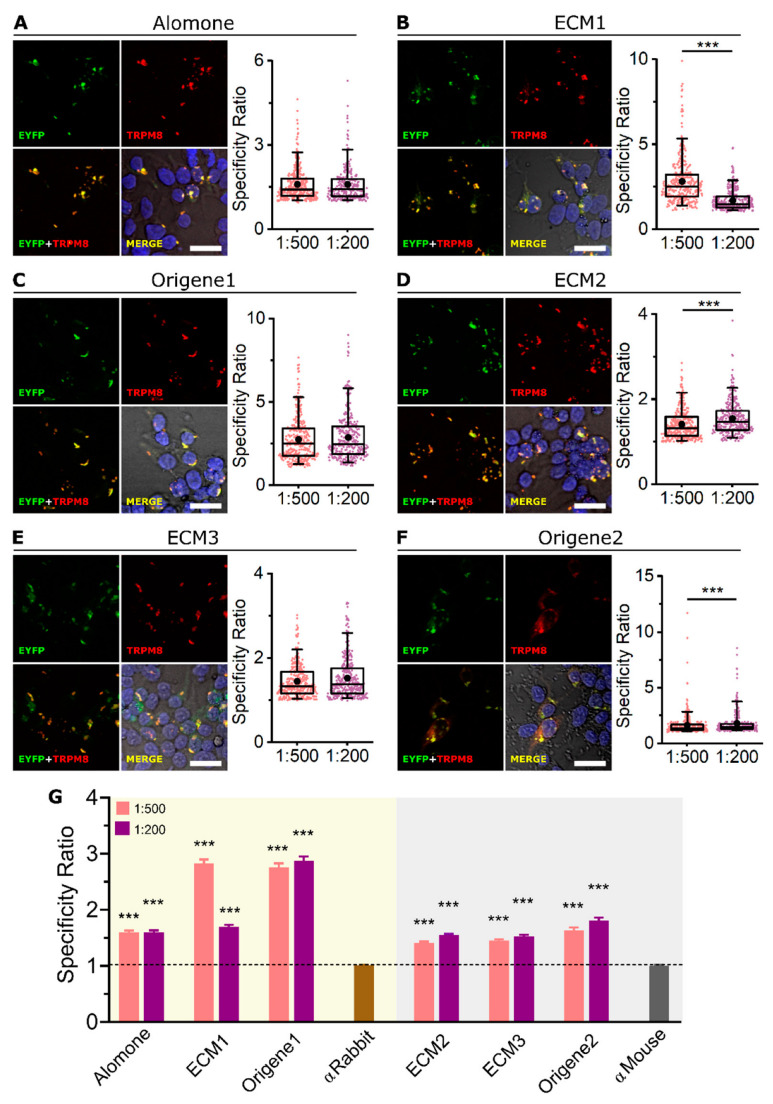
Immunocytochemistry of mTRPM8 in transfected HEK-293 cells. (**A**–**F** left) Confocal images of mTRPM8-EYFP transiently expressed in HEK-293 cells. EYFP (green), TRPM8 antibody (red) and Hoechst staining (blue). Merge images correspond to the overlap of the three fluorescent signals plus the bright field image. Scale bar: 30 µm. (**A**–**F** right) Box plots represent the specificity ratio (SR) for each dilution of the respective antibody. Each dot corresponds to an individual EYFP+ cell. Box contains the 25th to 75th percentiles. Whiskers mark the 5th and 95th percentiles. The line inside the box denotes the median and the black dot represents the mean. (*** *p* < 0.001, Mann–Whitney test). (**G**) Bar histogram summarizing the SR mean ± SEM of each TRPM8 antibody and the controls without primary antibody (αRabbit or αMouse secondary antibodies alone). Dashed line indicates SR = 1. (*** *p* < 0.001, Kruskal–Wallis followed by Dunn’s post hoc test vs. αRabbit or αMouse). For each antibody and dilution, n > 270 cells; 4 fields from 2 independent transfections.

**Figure 3 ijms-23-16164-f003:**
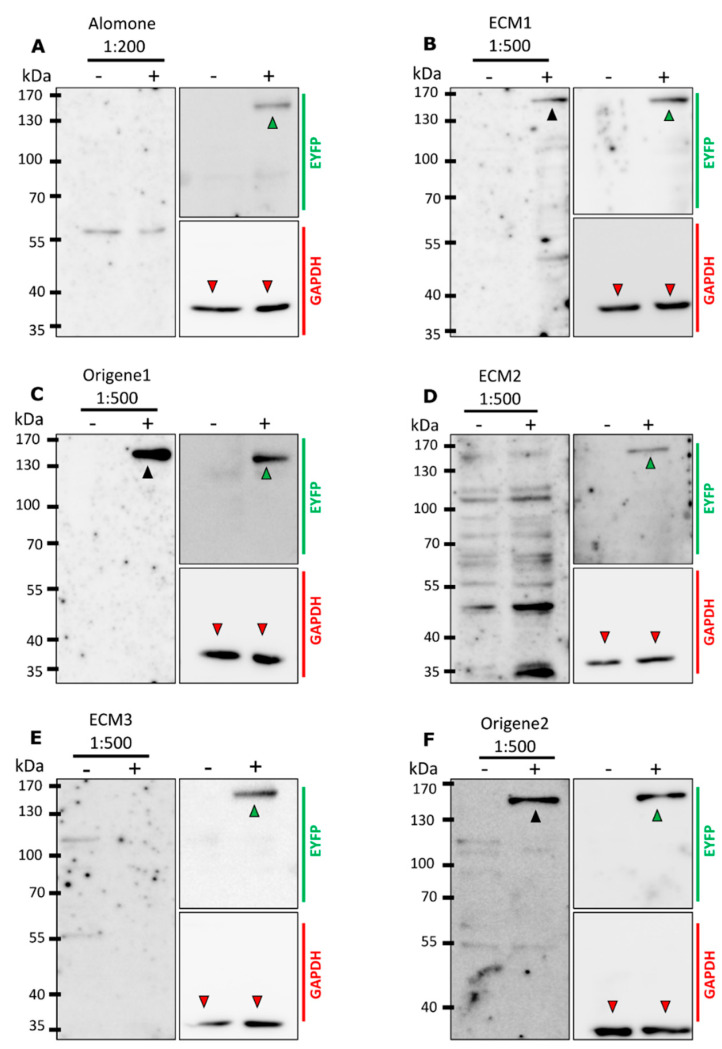
Western blot analysis for TRPM8 antibodies specificity. (**A**–**F**) TRPM8 immunoblots using (**A**) Alomone (**B**) ECM1, (**C**) Origene1, (**D**) ECM2, (**E**) ECM3 or (**F**) Origene2 antibodies. (−) lanes: untransfected HEK-293 cells, (+) lanes: HEK-293 cells transfected with mTRPM8-EYFP. Left: Immunoblot with each TRPM8 antibody. Right-top: EYFP immunoblotting on the same membrane. Right-bottom: GAPDH loading control. Black arrowheads indicate mTRPM8-EYFP bands revealed with antiTRPM8 antibody. Green arrowheads indicate mTRPM8-EYFP bands revealed with anti-GFP antibody. Red arrowheads indicate GAPDH bands. All blots were repeated at least 3 times to exclude a technical artefact when no anti-TRPM8 signal was observed. For each replicate, the same lysate was used for all antibodies.

**Figure 4 ijms-23-16164-f004:**
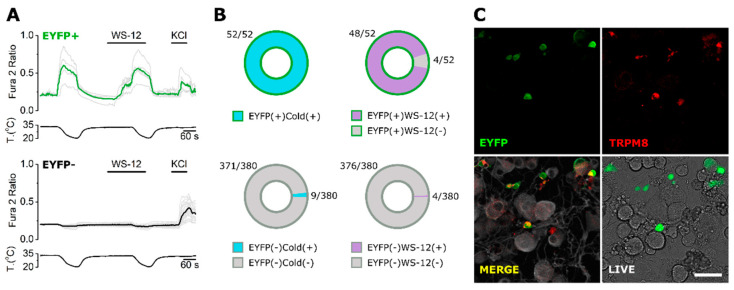
Calcium imaging in DRG-cultured cells from *Trpm8^BAC^-EYFP^+^* mice. (**A**) Representative traces of calcium transients evoked by mild cold (∼18–20 °C), WS-12 (10 µM) and KCl (30 mM) in EYFP+ and EYFP− neurons. Traces corresponding to individual cells are shown in grey. Colored traces represent averages of the respective individual traces. (**B**) Proportion of EYFP+ and EYFP− cells responding to cold and WS-12. (**C**) Representative immunocytochemistry performed after a calcium recording. Upper- and bottom-left: confocal images of the immunofluorescence. Green: EYFP. Red: TRPM8 (ECM1 antibody). Grey: βIII-Tubulin. Bottom-right: transmitted light and EYFP fluorescence from the same live cells before fixation. n = 432 cells from 2 mice. Scale bar: 50 µm.

**Figure 5 ijms-23-16164-f005:**
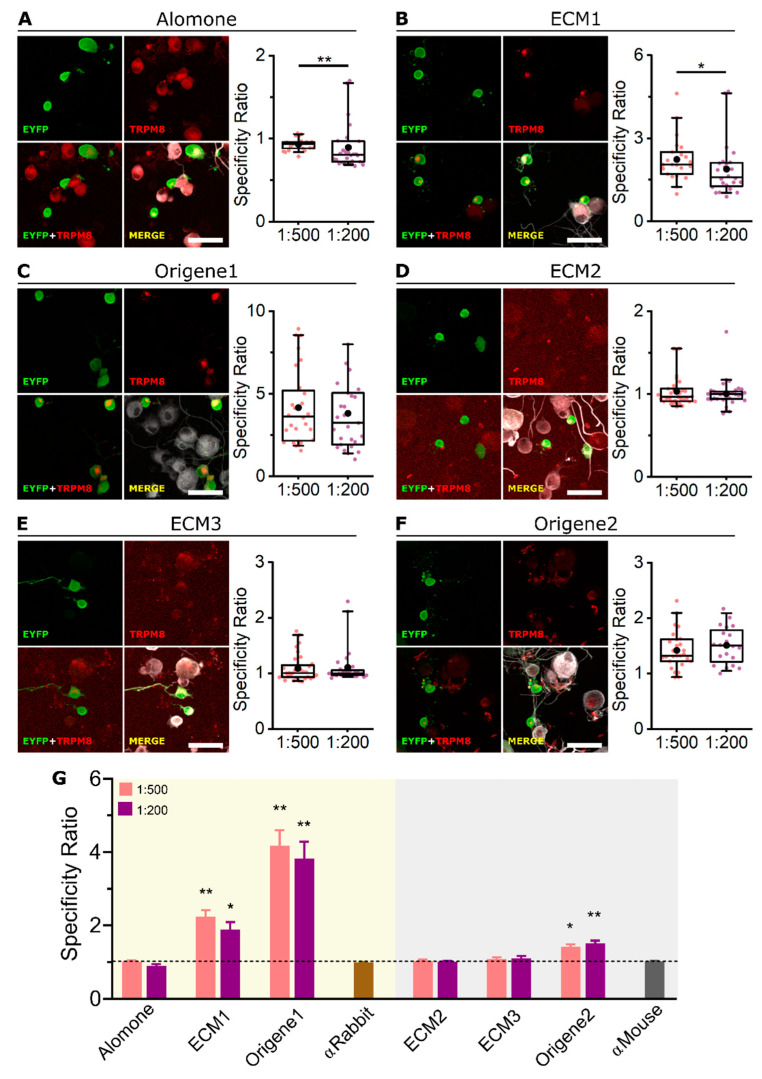
Immunocytochemistry of endogenous TRPM8 in cultured DRG cells from the *Trpm8^BAC^-EYFP^+^* mouse. (**A**–**F** left) Confocal images of TRPM8+ sensory neurons. EYFP (green), TRPM8 antibody (red) and βIII-Tubulin (grey). Scale bar: 50 µm. (**A**–**F** right) Box plots represent specificity ratio (SR) for each dilution of the respective antibody. Each dot corresponds to an individual EYFP+ cell. Box contains the 25th to 75th percentiles. Whiskers mark the 5th and 95th percentiles. The line inside the box denotes the median and the black dot represents the mean (* *p* < 0.05, ** *p* < 0.01, Mann–Whitney test). (**G**) Bar histogram summarizing the SR mean ± SEM of each TRPM8 antibody and the controls without primary antibody (αRabbit or αMouse secondary antibodies alone). Dashed line indicates SR = 1 (* *p* < 0.05, ** *p* < 0.01, Kruskal–Wallis followed by Dunn’s post hoc test vs. αRabbit or αMouse). n = at least 20 cells; 4 pictures from 2 mice for each antibody and dilution.

**Figure 6 ijms-23-16164-f006:**
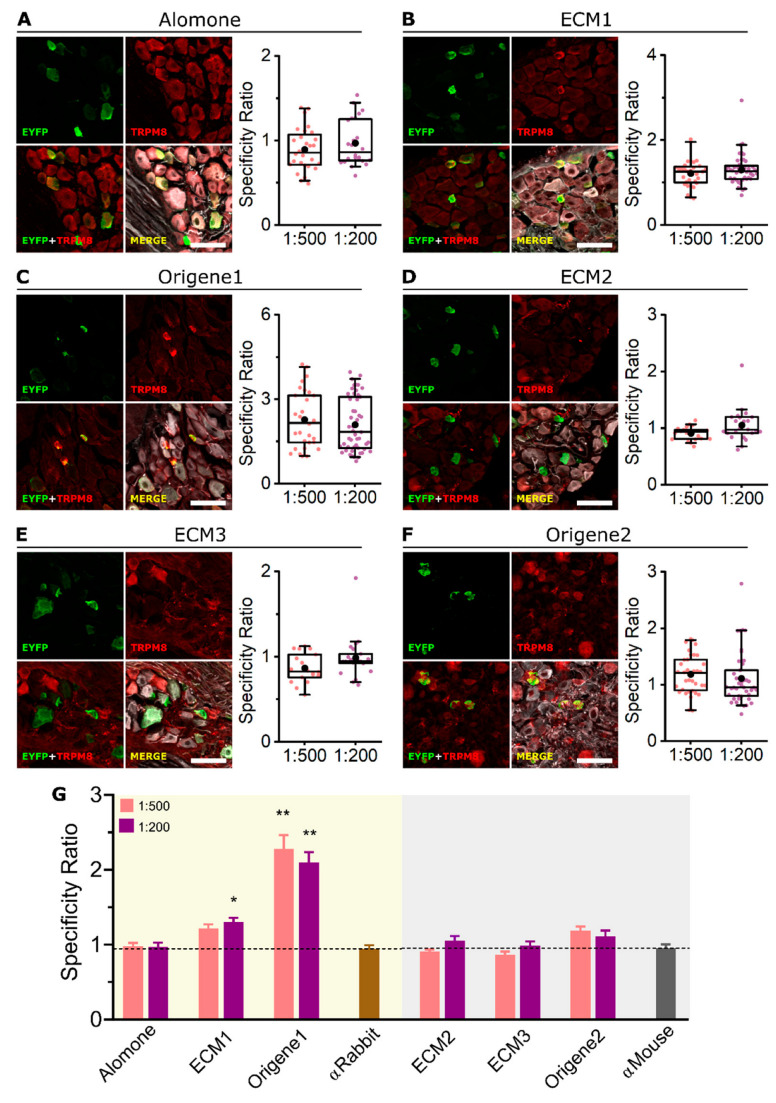
Immunohistochemistry of endogenous TRPM8 in DRG slices from the *Trpm8^BAC^-EYFP^+^* mouse. (**A**–**F** left) Confocal images of TRPM8+ sensory neurons. EYFP (green), TRPM8 antibody (red) and βIII-Tubulin (grey). Scale bar: 50 µm. (**A**–**F** right) Box plots represent specificity ratio (SR) for each dilution of the respective antibody. Each dot corresponds to an individual EYFP+ cell. Box contains the 25th to 75th percentiles. Whiskers mark the 5th and 95th percentiles. The line inside the box denotes the median and the black dot represents the mean. (Differences among dilutions were not significant, Mann–Whitney test). (**G**) Bar histogram summarizing the SR mean ± SEM of each TRPM8 antibody and the controls without primary antibody (αRabbit or αMouse secondary antibodies alone). Dashed line indicates SR = 1. (* *p* < 0.05, ** *p* < 0.01, Kruskal–Wallis followed by Dunn´s post hoc test vs. αRabbit or αMouse). n = at least 18 cells; 4 images from 2 mice for each antibody and dilution.

**Figure 7 ijms-23-16164-f007:**
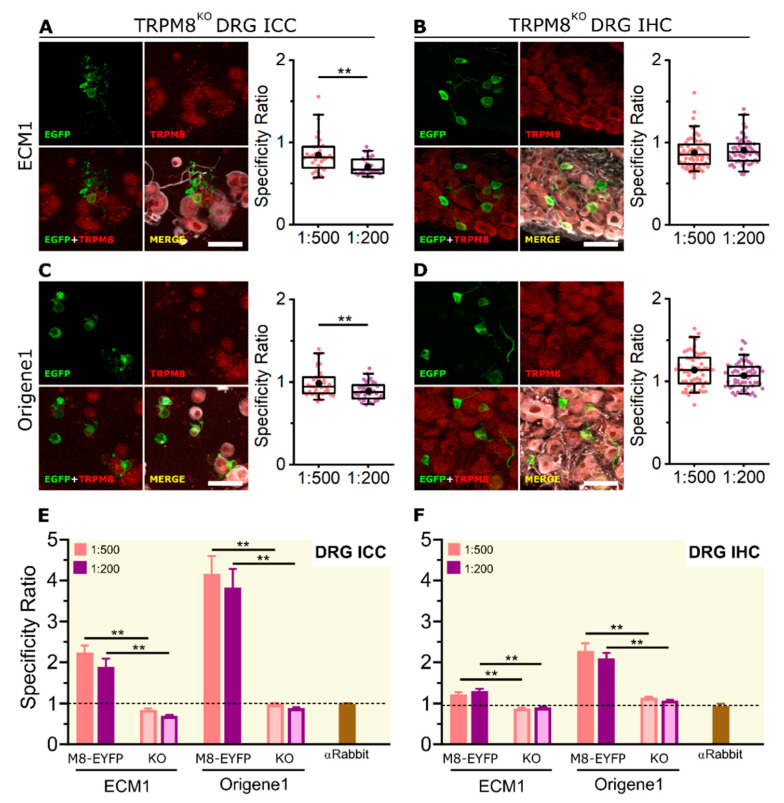
Immunofluorescence of endogenous TRPM8 in DRG cells and slices from the TRPM8 KO mouse. (**A**,**C**) Immunocytochemistry. (**B**,**D**) Immunohistochemistry. (**A**–**D** left) Confocal images of TRPM8 KO (*Trpm8^EGFPf^*; B6;129S1(FVB)-*Trpm8^tm1Apat^*/J) sensory neurons. EGFP (green), TRPM8 antibody (red) and βIII-Tubulin (grey). Scale bar: 50 µm. (**A**–**D** right) Box plots represent specificity ratio (SR) for each dilution of the respective antibody. Each dot corresponds to an individual EGFP+ cell. Box contains the 25th to 75th percentiles. Whiskers mark the 5th and 95th percentiles. The line inside the box denotes the median and the black dot represents the mean (** *p* < 0.01, Mann–Whitney test). (**E,F**) Bar histograms summarizing the SR mean ± SEM of each TRPM8 antibody in the KO and the reporter (*Trpm8^BAC^-EYFP^+^*, named M8-EYFP in the figure) mouse in ICC (**E**) and IHC (**F**). Dashed line indicates mean SR for the control without primary antibody (** *p* < 0.01, Mann–Whitney test). n = at least 22 cells; 4 pictures from 2 mice for each antibody and dilution.

**Table 1 ijms-23-16164-t001:** Antibody performance with the different techniques used in this study. − poor, + regular, ++ good, +++ excellent.

Antibody	Host	HEK-293 + TRPM8-EYFP		Mouse DRG	
		WB	ICC	ICC	IHC
Alomone	Rabbit	−	++	−	−
ECM1	Rabbit	+++	+++	++	++
Origene1	Rabbit	+++	+++	+++	+++
ECM2	Mouse	−	++	−	−
ECM3	Mouse	−	++	−	−
Origene2	Mouse	+++	++	+	−

**Table 2 ijms-23-16164-t002:** Antibodies used in this study.

Target	Name in This Study	Company	Ref.	Species	Use	Immunogen	Dilutions Used
TRPM8	Alomone	Alomone	ACC-049	Rabbit	Primary	Human TRPM8 917–929. Polyclonal	1:200 (WB, ICC, ICH) 1:500 (ICC, ICH)
TRPM8	ECM1	ECM biosciences	TP5701	Rabbit	Primary	Human TRPM8 extracellular domain. Polyclonal	1:200 (ICC, ICH) 1:500 (WB, ICC, ICH)
TRPM8	ECM2	ECM biosciences	TM5711	Mouse	Primary	Human TRPM8 extracellular domain. Monoclonal	1:200 (ICC, ICH) 1:500 (WB, ICC, ICH)
TRPM8	ECM3	ECM biosciences	TM5721	Mouse	Primary	Human TRPM8 extracellular domain. Monoclonal	1:200 (ICC, ICH) 1:500 (WB, ICC, ICH)
TRPM8	Origene1	OriGene	TA307827	Rabbit	Primary	Human TRPM8 extracellular domain. Monoclonal	1:200 (ICC, ICH) 1:500 (WB, ICC, ICH)
TRPM8	Origene2	OriGene	TA811228	Mouse	Primary	Human 1–300. Monoclonal	1:200 (ICC, ICH) 1:500 (WB, ICC, ICH)
GFP/YFP	EYFP	Abcam	ab13970	Chicken	Primary	GFP. Polyclonal	1:2000 (ICC, IHC)
GFP/YFP	EGFP	Invitrogen	A6455	Rabbit	Primary	Isolated directly from Jellyfish Aequorea Victoria Polyclonal	1:1000 (WB)
Class III β-tubulin	βIIItub-M	Biolegend	801201	Mouse	Primary	Rat brain microtubules. Monoclonal	1:1000 (ICC, IHC)
Class III β-tubulin	βIIItub-R	Biolegend	802001	Rabbit	Primary	Rat brain microtubules. Polyclonal	1:1000 (ICC, IHC)
GAPDH	GAPDH	Sigma	G9545	Rabbit	Primary	Synthetic peptide: residues 314–333 of mouse GAPDH	1:5000 (WB)
Mouse	Mouse HRP	Sigma	A9044	Rabbit	Secondary-HRP	Rabbit IgG fraction antiserum	1:4000 (WB)
Mouse	MouseA598	Invitrogen	A11005	Goat	Secondary-Alexa598	Mouse gamma globulin heavy and light chains	1:1000 (ICC, IHC)
Mouse	MouseA647	Invitrogen	A21237	Goat	Secondary-Alexa647	Mouse gamma globulin heavy and light chains	1:1000 (ICC, IHC)
Chicken	ChickenA488	Jackson IR	703-545-155	Donkey	Secondary-Alexa488	Chicken gamma globulin	1:1000 (ICC, IHC)
Rabbit	Rabbit HRP	Sigma	A9169	Goat	Secondary-HRP	Goat IgG	1:4000 (WB)
Rabbit	RabbitA555	Abcam	ab150062	Donkey	Secondary-Alexa555	N/A	1:1000 (ICC, IHC)

## Data Availability

Data are available from the authors on reasonable request.

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
