# Peer review of "Validation of Six Commercial Antibodies for the Detection of Heterologous and Endogenous TRPM8 Ion Channel Expression"

_ijms, 2022, doi:10.3390/ijms232416164_

Round 1
Reviewer 1 Report
The current study by Hernandez-Ortego and colleagues is a thorough appraisal of 6 commercial antibodies for the ion channel, TRPM8. The study is well conceived and has good controls, including comparing detection of TRPM8 in the DRG of wt vs TRPM8 KO mice. Studies of this kind are of value to the community and should help researchers in the field use the most appropriate antibody for the work they undertake. I have only a few minor points to make about this work:
1. Please correct the spelling errors. For example, there are several places where the manuscript has typos such as TRMP8.
2. The authors do a good job of describing their stats however, I think that they risk overstating statistical significance at times. This comes from the way that GraphPad / Prism calculates significance. Specifically, Prism models data and comes up with very small P values based on the data it has. This is not wrong per se but it is worth asking if it is really true that the authors actually reach P<0.0001 by studying 22 cells by immunofluorescence? I agree that they reach statistical significance but would argue that it is safer to say that the significance had a P value less than 0.01 (1%, or 1 in a hundred; rather than 1 in 10,000). This way, the authors do not suggest that they reached their P value in a manner that is statistically underpowered. In short, I think the data in this paper should be curtailed to *, P<0.05 and ** P<0.01.
Author Response
Thank you for your kind assessment of our manuscript and thank you for your useful suggestions.
- We have carefully read the manuscript and hope we have detected all, or at least most of the typos. We also fixed some punctuation errors.
- Regarding statistics, we agree with your view. There is really no need to overstate the significance with p values smaller than 0.01 for samples that are relatively small (i.e. neurons). In the case of samples of hundreds of cells (Figure 2), we think that the criterion could be relaxed to p < 0.001. We indicate this in the methods and corrected the figures and figure legends accordingly.
Reviewer 2 Report
Review report for the manuscript ijms 2099118
Proper detection and accurate quantification of TRPM8 using commercial antibodies are crucial for the quality and reliability of TRPM8 related studies. The manuscript by Pablo Hernandez-Ortego et al., “Validation of six commercial antibodies for detection of heterologous and endogenous TRPM8 ion channel expression” provides valuable reference by characterizing the performance of six TRPM8 commercial antibodies in three immunodetection techniques: western blot, immunocytochemistry and immunohistochemistry. Under the given conditions, they found that only two of six commercial TRPM8 antibodies being tested could successfully detect TRPM8 in all three approaches, while the other four were acceptable only for specific techniques. The data is solid and the manuscript is well written. I only have two minor suggestions listed below:
Line 127: “Hoescht” should be “Hoechst”.
Line 345: “performed” should be “perform”.

Author Response
Thank you for your kind assessment of our study, and also thank you for your suggestions. We carefully read the manuscript one more time, and corrected a few additional typos. We also corrected the errors you indicated in your report.
Line 127: “Hoescht” should be “Hoechst”. Fixed
Line 345: “performed” should be “perform". Fiexed